# Synthesis of Terminal-Alkylated PEGs with Imine Spacer to Form Iminium Mono-Ion Complexes for pDNA Delivery into Skeletal Muscle

**DOI:** 10.3390/pharmaceutics17081054

**Published:** 2025-08-13

**Authors:** Riku Oba, Yoko Endo-Takahashi, Yoichi Negishi, Shoichiro Asayama

**Affiliations:** 1Department of Applied Chemistry, Tokyo Metropolitan University, Hachioji, Tokyo 192-0397, Japan; oba-riku@ed.tmu.ac.jp; 2School of Pharmacy, Tokyo University of Pharmacy and Life Sciences, Hachioji, Tokyo 192-0392, Japan; endo@toyaku.ac.jp (Y.E.-T.); negishi@toyaku.ac.jp (Y.N.)

**Keywords:** plasmid DNA delivery, intramuscular injection, mono-ion complex, alkyl imine

## Abstract

**Background/Objectives:** To design the pDNA delivery carrier for delivery into skeletal muscle, a total of twelve terminal-alkylated PEGs (Cx-I-PEGy) with four alkyl groups of different carbon numbers (Cx: x = 4, 8, 12, 16) modified via an imine spacer at the ends of three methoxy PEGs of different molecular weights (PEGy: y = 500, 2k, 5k) have been synthesized. **Methods:** Among them, four Cx-I-PEG5k formed an imine-mediated complex formation with pDNA, as assessed by agarose gel electrophoresis, defined as an iminium mono-ion complex (I-MIC) without multivalent electrostatic interaction by minimizing potential toxic cations. **Results:** Most resulting I-MICs maintained the flexible structure of pDNA and promoted the binding to pDNA. The expression of pDNA by intramuscular injection with the resulting I-MICs was the highest by using I-MICs with C4-I-PEG5k and was observed extensively by the in vivo imaging system (IVIS). **Conclusions:** These results suggest that the I-MICs with C4-I-PEG5k are promising for pDNA transfection into skeletal muscle, offering the alkyl iminium for the pDNA binding group to demonstrate the factor of pDNA’s flexible structure as one of the key parameters for in vivo local pDNA transfection.

## 1. Introduction

pDNA encoding human hepatocyte growth factor (HGF) has shown significant activity in clinical trials for the treatment of peripheral arterial disease [1,2], and was first approved in Japan in 2019 for use by intramuscular injection. Thus, the intramuscular injection of pDNA could be an innovative approach in gene therapy. On the other hand, naked pDNA has difficulty approaching cells and uptaking into cells due to its rapid degradation by plasma nucleases and its huge molecular weight of 10^6^ Da [3]. Therefore, a carrier-based pDNA delivery system is necessary for the clinical application of pDNA as a therapeutic drug.

pDNA has negatively charged phosphate groups. Therefore, polyion complexes (PICs), which use a positively charged polymer (polycation) as a carrier and form a complex through electrostatic interaction between the polycation and pDNA, have been recognized as a useful approach in pDNA delivery. In general, modification of poly(ethylene glycol) (PEG) is used to control its cationic property in PICs (PEGylation). pDNA protection by PEG-modified amines minimizes nonspecific interactions with biological components, which are dispersed into the blood and provide stable circulation [4,5,6]. Furthermore, PEGylation improves diffusion-mediated permeation by minimizing electrostatic and hydrophobic interactions with the extracellular matrix [7].

However, the PEGylated PICs tend to exhibit their rigid structure from the multivalent electrostatic interaction of polycations with pDNA, so that the mono-ion complexes [8] (MICs) with mono-cation PEG as an alternative pDNA delivery strategy to polycations have been designed for their flexible structure. The electrostatic interaction between the alkyl imidazolium group and the phosphate group of pDNA was adjusted by the length of the alkyl chain to form a MIC with pDNA for in vivo gene transfection. Furthermore, ω-amide-pentylimidazolium end-modified PEG (APe-Im-PEG), which mimics DNA-binding proteins, stabilized MIC by forming hydrogen bonds between pDNA and the primary amide of APe-Im-PEG [9]. These MICs were less affected by multivalent electrostatic interactions such as PICs, and the increase in particle size was suppressed. This small particle size was suggested to be suitable for permeation via diffusion by PEGylation, facilitating in vivo pDNA transfection of the MICs.

Subsequently, our laboratory made the following improvements to the molecular design of MIC for efficient pDNA transfection in vivo: PEG is biocompatible and has improved permeability via diffusion, but its exclusion volume effect prevents its interaction with the cell membrane, and uptake is inhibited. This is called the PEG dilemma [10,11,12]. To solve this PEG dilemma and obtain highly efficient transfection of pDNA, we designed a mono-cation PEG in which the amide bond of the previous APe-Im-PEG was replaced by an ester bond [13,14]. A diethylamino end-modified PEG (DEAP-N = C-PEG) recently designed in our laboratory formed a more stabilized MIC due to the diethylamino group being a hydrophobic cation [15]. In DEAP-N = C-PEG, imine was selected as the spacer structure, which was able to dissociate PEG to solve the PEG dilemma, as well as the ester bond. However, compared to previous mono-cation PEG with amide or ester spacer, the imine showed higher complex formation ability and higher stability in the presence of serum, suggesting its usefulness. The molecular design of mono-cation PEG used in previous MICs indicates that the design of spacer structures such as amides, esters, and imines, as well as terminal functional groups that interact electrostatically with pDNA, are important factors for MIC formation and pDNA transfection.

In this study, for the establishment of a unique conceptualization for pDNA delivery carrier design with higher MIC formation ability and in vivo pDNA transfection activity, a terminal-alkylated PEG with imine spacer (Cx-I-PEGy) has been synthesized. Since it has been suggested that hydrophobic properties are involved in gene transfection [16,17,18], linear alkyl chains were selected as the terminal modification groups, with the four types being butyl (C4), octyl (C8), dodecyl (C12), and hexadecyl (C16). The imine was intended to act as a point of electrostatic interaction with pDNA, resulting in the iminium mono-ion complex (I-MIC) defined in this study. The I-MIC is expected to have a nanoscale, with its flexibility preserving pDNA structure by minimizing potential toxic cations, as compared to other non-viral carriers. One imine per carrier, because of the MIC and not PIC, is a weak base, which is considered to work as a bio-safe non-ion under physiological pH without cytotoxic cations. When the I-MICs are formed, the imine is induced to be cationic iminium, expecting the minimization of the potential toxic cations under physiological pH. The physicochemical properties were verified by dynamic light scattering (DLS) and circular dichroism (CD) measurement, as well as transmission electron microscopy (TEM) observation. The PEG was used with molecular weights of approximately 500 (PEG500), 2000 (PEG2k), and 5000 (PEG5k) Da. A total of twelve carriers with these four alkyl chains and three PEG chains via an imine spacer have been synthesized and assessed for both the complex formation with pDNA and the transfection efficiency to explore useful carriers for pDNA skeletal muscle delivery.

## 2. Materials and Methods

### 2.1. Materials

α-methoxy-ω-(3-oxopropoxy) and polyoxyethylene (mPEG-CHO) with molecular weights of 550 and 2000 Da were purchased from Creative PEGWorks (Research Triangle Park, NC, USA). α-methoxy-ω-(3-oxopropoxy) and polyoxyethylene (mPEG-CHO) with molecular weights of 5000 Da was purchased from NOF CORPORATION (Tokyo, Japan). n-Octylamine, n-dodecylamine, and n-hexadecylamine were purchased from Tokyo Chemical Industry Co., Ltd. (Tokyo, Japan). n-Butylamine, molecular sieves 3 A 1/8 (MS), and chloroform were purchased from FUJIFILM Wako Pure Chemical Corporation (Osaka, Japan). *N*,*N*-dimethylformamide (DMF), and hexane were purchased from Kanto Chemical Co., Inc. (Tokyo, Japan). All other chemicals of a special grade were used without further purification.

### 2.2. Synthesis of Cx-I-PEGy

The procedure to synthesize Cx-I-PEGy is as follows: mPEG-CHO of molecular weights of 550, 2000, and 5000 Da and 5-fold molar amounts of each alkylamine, such as n-butylamine (C4), n-octylamine (C8), n-dodecylamine (C12), and n-hexadecylamine (C16), to mPEG-CHO and MS were mixed in 2 mL of DMF, followed by stirring at 40 °C for up to 72 h. After the stirring, the MS was removed by filtration with 0.22 µm pore diameter poly(tetrafluoroethylene), followed by evaporating the mixture to remove the DMF. The resulting mixture was dissolved in 1 mL of chloroform, and the solution was added to 20 mL of hexane to remove unreacted alkylamines. Then, the precipitate in the solution was collected by centrifugation, followed by vacuum-drying to obtain the product.

### 2.3. ^1^H NMR Spectroscopy

Synthesized Cx-I-PEGy (3 mg) was dissolved in 500 μL of CDCl_3_ (99.8 atom% deuterium, 0.03% TMS, and containing silver foil; Kanto Chemical Co., Inc., Tokyo, Japan). The ^1^H NMR spectra (500 MHz) were obtained on a Bruker AV500 spectrometer (Billerica, MA, USA).

### 2.4. Agarose Gel Electrophoresis to Confirm the I-MIC Formation

A stock water solution of various Cx-I-PEGy and pDNA stock solutions (300 ng pDNA/sample) was mixed with water, where the final volume was adjusted to 13.5 μL at various mixing ratios, [Imine]_Cx-I-PEGy_/[Phosphate]_pDNA_ (N/P), followed by incubation at 37 °C for 24 h. For the DNA marker, TrackItTM 1 Kb Plus DNA Ladder (100–15,000 bp), 500 ng DNA/sample, and water were mixed, where the final volume was adjusted to 12.5 μL. After mixing with a loading buffer (1.5 μL) (For DNA markers, 2.5 µL of loading buffer was mixed), the resulting sample was loaded onto a 1% agarose gel containing 1 μg/mL ethidium bromide. Gel electrophoresis (50 V, 30 min) was performed at room temperature in a TAE buffer (Tris-acetate, EDTA), followed by the visualization of the pDNA bands under UV irradiation. For the salt resistance assay of the complexes, the electrophoresis was performed after the complexes at the mixing ratios (N/P) of 1 and 16 were incubated at 37 °C for 30–240 min in phosphate buffered saline (PBS). For the polyanion resistance assay of the complexes, the electrophoresis was performed after the complexes at the mixing ratios (N/P) of 1 and 16 were incubated at 37 °C for 30–240 min in the presence of 8 mM [SO_3_^−^] dextran sulfate (DS). For the stability assay of the complexes, the electrophoresis was performed after the complexes at the mixing ratios (N/P) of 1 and 16 were incubated at 37 °C for 30–240 min in the presence of 10% fetal bovine serum (FBS).

### 2.5. Critical Micelle Concentration (CMC) Measurement

Cx-I-PEGy stock solutions in water with various concentrations were prepared, followed by adding 10 µL of the pyrene stock solution (0.5 mg/mL) prepared with acetone. Fluorescence intensity was measured from 360 nm to 450 nm (excitation wavelength: 339 nm) using a spectrofluorometer. The vertical axis represents the ratio of intensity of the first (I_1_) and third peak (I_3_). The horizontal axis represents the log concentration of each Cx-I-PEGy [19,20].

### 2.6. Acid–Base Titration

The C8-I-PEG5k solution (2.4 mM) was prepared. The 1 mol/L sodium hydroxide (NaOH) solution (FUJIFILM Wako Pure Chemical Corporation, Osaka, Japan) was added to each solution to allow the solution to reach pH 12. After adding NaOH, the 1 mol/L hydrochloric acid (HCl) (FUJIFILM Wako Pure Chemical Corporation, Osaka, Japan) was added in 2 µL increments to reach pH 11. Then, 0.1 mol/L HCl was added in 2 µL increments to reach pH 2. During HCl addition, the pH of the solution was measured with a pH meter (HORIBA Scientific, Kyoto, Japan).

### 2.7. Particle Size and Zeta (ζ) Potential Measurement

A dynamic light scattering (DLS) method by an electrophoresis light scattering spectrophotometer (ELSZneo, Otsuka Electronics Co., Ltd., Tokyo, Japan) determined the size of the pDNA (5 μg), which was incubated at 37 °C for 24 h with the Cx-I-PEGy at the mixing ratios (N/P) of 1 and 16 in 100 μL of water (*n* = 5). The zeta potential of the resulting sample (10 × diluted) was measured by an electrophoretic light scattering (ELS) method using the same instrument (*n* = 3).

### 2.8. Circular Dichroism (CD) Spectrum Measurement

After the complex of pDNA (5 µg) with each Cx-I-PEGy was formed, the solution volume was adjusted up to 350 μL with water, and the CD spectrum from 230 nm to 310 nm was measured with a circular dichroism spectropolarimeter (J-820, JASCO Co., Ltd., Tokyo, Japan).

### 2.9. Transmission Electron Microscopy (TEM) Observation

The TEM observation was carried out according to the previous literature [21]. The TEM sample solution for observing the complex was prepared by mixing 2 μL of a twice-diluted complex solution with 2 μL of 2% uranyl acetate on ice. The TEM grid (Nisshin EM Co., Tokyo, Japan), which had been hydrophilized by an Eiko IB-3 ion coater (Eiko Engineering Co., Ltd., Shimane, Japan), was dipped into the sample solution for 45 s. The excess solution was blotted away. The grids were observed by a JEM-1400 Bio-TEM (JEOL Ltd., Tokyo, Japan) operated at an acceleration voltage of 120 kV.

### 2.10. Cell Viability Assay

As a representative cell, mouse myoblast cell line C2 C12 cells (from Riken Bioresource Center Cell Bank) were cultured in tissue culture flasks in Dulbecco’s modified Eagle’s medium (DMEM) supplemented with 10% heat-inactivated FBS. The cells were seeded at 1 × 10^4^ cells/well in a 96-well plate, followed by incubation overnight at 37 °C in a 5% CO_2_ incubator. Then, the cells were treated with each Cx-I-PEG5k/pDNA complex, followed by further incubation for 24 h at 37 °C. By additional incubation for 4 h, the cell viability was measured by the alamar-blue assay (*n* = 3) [22].

### 2.11. Hemolysis Assay

The 164 µL of 10 mM phosphate buffer (PB) (pH 7.4 or 5.1) containing 130 mM NaCl and 16 µL of Cx-I-PEG5 k solution (0.64, 6.4, 64 mg/mL) was mixed. For branched poly(ethylenimine) (bPEI), the 179 µL of that buffer and 1 µL of bPEI solution (0.20, 2.0, 20 mg/mL) were mixed. The mixture was incubated with 20 µL of mouse preserved blood for 24 h at 37 °C. After centrifugation (13,200 rpm, 10 min, 4 °C), the amount of released hemoglobin from erythrocytes was determined by measuring the absorbance of the 5-fold diluted supernatant at 570 nm (*n* = 6) from heme using a microplate reader [23,24].

### 2.12. In Vitro Gene Transfection Activity

In a typical 96-well plate experiment, 1 × 10^4^ cells/well C2C12 cells were transfected in DMEM supplemented with 10% heat-inactivated FBS by the addition of 10 μL of PBS(−) containing 300 ng of pcDNA3-Luc plasmid complexes with Cx-I-PEG5 k. The pcDNA3-Luc plasmid was derived from pGL3-basic (Promega, Madison, WI) and used as a plasmid DNA encoding the firefly luciferase gene under the control of a cytomegalovirus promoter. As a positive control, bPEI was used (charge ratio (+/−) = 1, 8, 16). After 2 days of incubation, the cells were subjected to the luciferase assay (Promega kit) according to the manufacturer’s instructions. Luciferase activities were normalized by protein concentrations and are represented as relative light units (RLUs) (*n* = 3). Protein concentrations were determined by BCA protein assay kit (Pierce) according to the manufacturer’s instructions (*n* = 1).

### 2.13. In Vivo Gene Transfection Activity

In vivo gene delivery to the skeletal muscles of mice with each Cx-I-PEG5k was carried out using previously described methods [25]. Briefly, ICR mice (five weeks old, male) were anesthetized with pentobarbital. The complexes of pDNA (5 µg) with each Cx-I-PEG5 k at the mixing ratio (N/P) of 1 and 16 in 26.3 µL of water were incubated at 37 °C for 24 h, followed by adding to the mixture 8.7 µL of 4 × PBS. Then, the complexes were injected into the tibialis muscles of the ICR mice. One week after the injection, the whole tibialis muscles were collected and homogenized in a lysis buffer (0.1 M Tris-HCl: pH 7.8, 0.1% Triton X-100, and 2 mM EDTA). Luciferase activity was measured with a luminometer (LB96 V, Belthold Japan Co. Ltd., Tokyo, Japan) according to a luciferase assay system (Promega, Madison, WI, USA). The luciferase activity is normalized by relative light units (RLUs) per mg of protein. The pDNA used for the in vitro assessment was used.

### 2.14. In Vivo Imaging System (IVIS)

Luciferase gene expression was observed using an in vivo imaging system (IVIS Lumina Series III, PerkinElmer, Waltham, MA, USA) for one week. During the measurement, VivoGro Luciferin (Promega; 10 μg per mg body weight) was administered intraperitoneally; the samples were incubated for 10 min before observation.

### 2.15. Animals

The use of animals and relevant experimental procedures were approved by the Tokyo University of Pharmacy and Life Science Committee on the Care and Use of Laboratory Animals. Approval Code: P23-65, P24-53, P25-60. Approval Date: 7 May 2024 (P25-60); 16 May 2024 (P23-65, P24-53).

### 2.16. Statical Analysis

Statical analysis was performed using the two-sample equal variance Student’s *t*-test (Microsoft 365 Excel).

## 3. Results and Discussion

### 3.1. Synthesis and Screening by pDNA Complex Formation Ability of Cx-I-PEGy

Figure 1 shows a scheme of the Cx-I-PEGy synthesis. This chemical reaction is a reversible reaction in which a primary amine makes a nucleophilic attack on the carbonyl carbon of mPEG-CHO to produce H_2_O. A total of twelve carriers were synthesized with four alkylamines (Cx: C4, C8, C12, C16) and three mPEGs-CHO (PEGy: PEG500, PEG2k, PEG5k: without terminal -CH_2_CH_2_CHO). The same synthetic method was used for all carriers. First, each alkylamine was reacted with each mPEG-CHO in DMF solvent, where molecular sieve (MS) was mixed to remove the water byproduct. Filter filtration was used to remove MS, and evaporation was used to remove DMF. Then, the synthesis was completed by purification with chloroform and hexane to remove unreacted alkylamines. The results of the ^1^H NMR spectra of the obtained product (Cx-I-PEGy) and each of the reactants (four alkylamines and three mPEGs-CHO) are shown in Appendix A. Successful purification was confirmed by the disappearance of the proton signal due to the aldehyde of mPEG-CHO (signal CHO: Appendix A) and the disappearance of the proton signal due to the methylene group adjacent to the primary amine of the unreacted alkylamine (signal g: Appendix A). Although ^13^ C-NMR and HR-MS data is preferable for full characterization, it is difficult to detect a terminal small alkyl group in a huge PEG macromolecule. Therefore, for each Cx-I-PEGy, successful synthesis was confirmed by the ratio of the proton signal due to the methoxy group of mPEG-CHO (signal a: 3 H) to the proton signal due to the terminal methyl group (signal i: 3 H), which was approximately 1:1.

Agarose gel electrophoresis was performed to screen the ability of each of the twelve resulting Cx-I-PEGy to form pDNA complexes, and the results are shown in Figure 2. Three PEG molecular weights (500, 2k, 5k) were tested because 5k was used as a standard for biodistribution/permeability [7], and 500 and 2k were used as controls. The lower ability of Cx-I-PEG500 and Cx-I-PEG2k to form complexes at all alkyl chain lengths was observed. In contrast, Cx-I-PEG5k formed complexes despite a lower mixing ratio ([Imine]_Cx-I-PEGy_/[Phosphate]_pDNA_: N/P), and the complex formation was enhanced in dependence on increasing N/P. The ability of Cx-I-PEG5k to form complexes increased with increasing alkyl chain length in the context of N/P ratio. At a N/P ratio of one, especially, it is clear that the retardation of the pDNA band increased with increasing alkyl chain length. At a N/P ratio of 16, especially, no free pDNA band was observed in the presence of all Cx-I-PEG5k, suggesting 100% MIC formation. This is due to the fact that the longer alkyl chains increased the hydrophobicity of the carrier, resulting in stronger binding to the hydrophobic moieties of the pDNA base pair. The increase in the hydrophobicity of Cx-I-PEGy was supported by the assessment of the hydrophobic field with pyrene [19,20] (Appendix A). The I_1_/I_3_ ratio is helpful for determining the location of the pyrene probe in the micelles. The I_1_/I_3_ value of approximately 1.0 indicates that the pyrene probe is placed in the alkyl chain core region in the micelles. On the other hand, a value of approximately 1.8 indicates that no micelle formation places the pyrene. In the case of PEG500 and PEG2k, the longer the alkyl chain length, the lower I_1_/I_3_ values (approximately 1.0), suggesting more aggregation (micelle formation) occurs (Appendix A). On the other hand, in the case of PEG5k, higher I_1_/I_3_ values (approximately 1.8) were estimated in spite of a hexadecyl terminal (C16) (Appendix A), suggesting no interaction (no micelle formation) between Cx-I-PEG5k polymers. Therefore, Cx-I-PEG5 k easily interacted with pDNA and formed a complex. Each reactant mPEG-CHO (M.W. 500, 2000, 5000) is unlikely to form pDNA complexes (Appendix A). And each reactant alkylamine (C4, C8, C12, and C16) is insoluble in water, so the complex formation ability was not able to be assessed. Notably, Cx-I-PEG5k also forms complexes with mRNA (Appendix A). These results suggest that Cx-I-PEG5k has formed a pDNA and mRNA complex via terminal-alkylated iminium. In subsequent studies, Cx-I-PEG5k, for which significant complex formation was observed, was used for assay. The ideal mixing N/P ratio of 1 and the N/P ratio of 16 (an excess amount of Cx-I-PEG5k) were selected for the subsequent assay.

### 3.2. How to Form pDNA Complex with Cx-I-PEG5k

In the previous section, Cx-I-PEG5k was found to form pDNA complexes in water. Therefore, the way in which Cx-I-PEG5k forms pDNA complexes was investigated by changing the solvent. The Cx-I-PEG5k and pDNA were mixed in phosphate buffer (PB) at pH 6, 7, and 8. The ability to form complexes was evaluated by agarose gel electrophoresis, and the results are shown in Figure 3. At pH 8 and pH 7, less retardation of pDNA bands was observed, suggesting that the ability of each Cx-I-PEG5k to form complexes was low. On the other hand, at pH 6, more retardation of pDNA bands was observed, suggesting that all alkyl chain lengths of Cx-I-PEG5k formed pDNA complexes. This is due to the fact that the pK_a_ of imine in the carrier structure is less than 7. As shown in Figure 4, acid–base titration of C8-I-PEG5k as a representative carrier exhibited proton buffering capacity below pH 7. The resulting acid–base titration curve in the range from pH 11 to pH 3 by adding HCl suggests that the imine was not protonated above pH 7, and interaction with the anionic pDNA did not occur. Since this complex was formed in water with no pH buffering capacity (Figure 2), a sufficient complex of Cx-I-PEG5k must be formed in water. Conversely, from no retardation of pDNA bands (Appendix A), the complex was not formed in phosphate buffered saline (PBS) as the solvent during complex formation. This is presumably due to the loss of positive charge in PBS at pH 7.4 with abundant salts under physiological ionic strength conditions. These results suggest that Cx-I-PEG5k forms a complex with pDNA by electrostatic interaction, due to protonation of some imines in water, without pH buffering capacity. Thus, we have called the resulting complex “Iminium Mono-Ion Complex (I-MIC)”. Therefore, water was selected as the solvent for the I-MIC formation of Cx-I-PEG5k and pDNA in subsequent assessments.

### 3.3. Physicochemical Properties of the Cx-I-PEG5k/pDNA I-MICs

The physicochemical properties of the I-MICs were assessed. The CD spectrum of pDNA in the I-MICs by Cx-I-PEG5 k is shown in Figure 5. At a mixing N/P ratio of 1, the peak was not shifted compared to the naked pDNA spectrum, suggesting that the conformation of pDNA was not changed. On the other hand, at a mixing N/P ratio of 16, except for C4-I-PEG5k, a long wavelength shift and a decrease in the molar ellipticity of the positive peak were observed, suggesting that the conformation of pDNA was a little changed. From the agarose gel electrophoresis, fluorescence from ethidium bromide was observed in the Cx-I-PEG5k/pDNA I-MICs despite the retardation of pDNA bands (Figure 2). The resulting fluorescence indicates that ethidium bromide intercalated with pDNA, suggesting partial condensation of naked pDNA, which was consistent with a little conformation change of pDNA. In case of a control PIC, complete condensation with no fluorescence from ethidium bromide exhibited a change in pDNA conformation [26]. No condensation, such as naked pDNA, is the significance of pDNA conformation in gene delivery for transcription and translation. Furthermore, as shown in Figure 6, TEM images of a representative I-MIC with a size distribution indicated by the arrows, C4-I-PEG5k/pDNA I-MIC (N/P = 16), were also shown. Based on contrast alone, the resulting structures became less dark after the addition of C4-I-PEG5k, which was not observed under these experimental conditions, indicating an interaction of the C4-I-PEG5k with pDNA. The resulting sizes seem comparable, probably due to no observation of the C4-I-PEG5k. Comparing naked pDNA with the I-MIC, no significant change in its morphology was observed. Taking these results into account, the structure of pDNA in the I-MICs is presumable flexible as naked pDNA. The presumable flexible structure is expected to lead to improved permeability in the tissues.

The particle size and zeta potential of each Cx-I-PEG5k/pDNA I-MIC are shown in Figure 7. The pDNA I-MICs with Cx-I-PEG5k exhibited a particle size of 100–300 nm with a zeta potential of medium anionic to small cationic. Especially, the observed positive zeta potential values, combined with the CD signal shift in the case of C16-I-PEG5k without the background signal, may be attributed to an excess positive charge and an excess of polymer relative to pDNA. Also, the I-MICs exhibited a polydispersity index (PDI) of 0.15–0.30 (Appendix A), which was almost consistent with the particle size distribution curves (Appendix A). Each zeta potential of the I-MICs was almost consistent with the results of agarose gel electrophoresis (Figure 2), which assessed the ability of the I-MICs’ formation, because the net negative zeta potential (except at high N/P ratios) was generally near neutral, as compared to the values of the naked pDNA. From the generally anionic nature of the Cx-I-PEG5k/pDNA I-MICs, it is expected to reduce nonspecific interactions with biomolecules such as serum proteins in vivo. Especially, in spite of a high N/P ratio of 16, the zeta potential of the C4-I-PEG5k/pDNA I-MICs was negative, suggesting the suitability of the I-MIC for in vivo therapeutic use.

### 3.4. Stability of Cx-I-PEG5k/pDNA I-MICs

Although the I-MICs have been formed in water, the in vivo environment is more complicated than in water. Therefore, the formation of the Cx-I-PEG5k/pDNA I-MICs was assessed under physiological ionic strength conditions using PBS, because the stability of the I-MICs, formed by electrostatic interaction, depended on ionic strength. The Cx-I-PEG5k/pDNA I-MICs were formed in water, followed by the addition of PBS. In all Cx-I-PEG5k/pDNA I-MICs, the pDNA bands were further migrated at 30 min after the addition of PBS, and the migrated bands were retained until 240 min (Appendix A). These results suggest that the I-MICs were partially dissociated in PBS, presumably due to the loss of positive charge in PBS at pH 7.4 under physiological ionic strength conditions (Appendix A). However, the alkyl chains of Cx-I-PEG5k may contribute to form the I-MIC with pDNA by hydrophobic interaction.

Then, the stability of the Cx-I-PEG5k/pDNA I-MICs was assessed by competitive exchange with polyanions such as DS [27,28]. In all Cx-I-PEG5k/pDNA I-MICs, the change in the bands was slight (Appendix A), suggesting the negligible dissociation of I-MICs. The resulting stability of the I-MICs against competitive exchange with polyanions in spite of the mono-cation Cx-I-PEG5k supports the hydrophobic interaction between pDNA and the alkyl chains of Cx-I-PEG5k.

Finally, to examine the hydrophobic interaction for the I-MIC formation, as shown in Figure 8, we assessed the stability of the Cx-I-PEG5k/pDNA I-MICs in the presence of serum proteins such as fetal bovine serum (FBS), including DNase. In all Cx-I-PEG5k/pDNA I-MICs, the pDNA bands were retained at 30 min after the addition of FBS. Especially, at mixing ratios (N/P) of 16, almost no change of the pDNA bands in the I-MICs at 30 min suggests negligible protein corona formation and aggregation in serum. Notably, the pDNA bands in the I-MICs using Cx-I-PEG5k with longer alkyl chains were retained for a longer time, until 120 min. These results suggest that more hydrophobic interaction with the longer alkyl chains of Cx-I-PEG5k stabilized the formation of the I-MICs. Namely, the hydrophobic interaction between pDNA and the alkyl chains of Cx-I-PEG5k is considered to be a key factor for the I-MIC formation. Thus, no significant dissociation of the Cx-I-PEG5k/pDNA I-MICs in the presence of FBS is expected to be useful in vivo.

### 3.5. Pharmaceutical Properties of Cx-I-PEG5k/pDNA I-MICs In Vitro

The biochemical properties of the Cx-I-PEG5k/pDNA I-MICs in vitro were evaluated. The cytotoxicity of the Cx-I-PEG5k/pDNA I-MICs was assessed. Since bPEI is considered the gold standard as a pDNA delivery carrier in vitro [29,30], it was used as a control in the cytotoxicity assay. The bPEI is a polycation from a PIC with higher cytotoxic cation density than linear PEI, and lipofectamine is a lipid assembly, so we used bPEI as a positive control to compare I-MICs with a representative PIC. The cells used were C2C12 mouse myoblasts derived from muscle satellite cells. As shown in Figure 9, cell viability decreased with longer alkyl chains in the Cx-I-PEG5k/pDNA I-MICs at the mixing N/P ratio of 16. At the mixing N/P ratio of 1, on the other hand, cell viability did not decrease, even in case of C16-I-PEG5k with the longest alkyl chain. We did not further analyze the trend of cell viability with the increase of alkyl chain length at the mixing N/P ratio of 1, because the C16-I-PEG5k/pDNA I-MIC exhibited the apparent different (large) particle size as compared to other I-MICs (Figure 7A). The hydrophobic and hydrophilic portions of Cx-I-PEG5k as our pDNA delivery carrier exhibited a surfactant effect, suggesting that they easily disrupted the cell membrane and caused cytotoxicity. The intrinsic cytotoxicity of 1 mM Cx-I-PEG5k is estimated from hemolytic assay at pH 7 (Appendix A). The longer the alkyl chain of the Cx-I-PEG5 k, the higher the hemolytic activity at pH 7 (Appendix A). The resulting hemolytic activity induced by the Cx-I-PEG5k with longer alkyl chains supports the results of the cytotoxicity assay (Figure 9). Especially, the cell viability at the mixing N/P ratio of 16 for C16-I-PEG5k was approximately 30%, lower than bPEI, therefore suggesting that the surfactant effect stood out from the PEG shielding effect.

pDNA transfection efficiency in vitro was higher in conditions with longer alkyl chains as compared to naked pDNA (Appendix A, left axis). In the protein determination, the I-MICs with a high mixing ratio (N/P = 16) show lower similar values (lower cell viability) compared to the bPEI/pDNA complex with a high mixing charge ratio (+/− = 16) (Appendix A, right axis). Neither cellular uptake nor endosomal escape mediated gene transfection in vitro, although direct mechanistic experiments are necessary, so these results suggest that the surfactant effect of Cx-I-PEG5k with longer alkyl chains disrupted the cell membrane (lower cell viability) to enhance cell uptake (higher gene transfection efficiency). Our previous report shows that the relationship between in vivo and in vitro gene transfection efficiency does not correlate [31], while the reason why naked pDNA showed comparatively high gene expression after intramuscular injection is unknown in spite of discussion [32]. Therefore, all of the Cx-I-PEG5k were used in subsequent in vivo gene transfection.

### 3.6. Pharmaceutical Properties of Cx-I-PEG5k/pDNA I-MICs In Vivo

In vivo gene transfection efficiency of the Cx-I-PEG5k/pDNA I-MICs was assessed. The Cx-I-PEG5k/pDNA I-MICs were injected into the tibialis muscle of mice. As shown in Figure 10, the efficiency of in vivo gene expression mediated by each Cx-I-PEG5k with a different alkyl chain length (Figure 10A–C) after one week was assessed by luciferase reporter assay. In vivo gene expression is greatly affected by individual differences of mice (almost *n* = 4). Furthermore, different lots of naked pDNA significantly change the gene expression level. As far as we know, there is no report of transfection efficiency mediated by PEG-based PIC micelles. Therefore, we normalized the gene expression levels of each I-MIC by dividing the analysis for each experiment with different naked pDNA and calculating the relative gene expression efficiency of each Cx-I-PEG5k/pDNA I-MIC to the gene expression efficiency of the naked pDNA (Figure 10D). As a result, the C4-I-PEG5k/pDNA I-MIC (N/P = 16) mediated the highest gene expression, as compared to other I-MICs and bPEI (representative PIC), where the comparison to other clinically relevant gene delivery systems, such as lipid nanoparticles and viral vectors, was of a future scope. Although partial condensation of naked pDNA (Figure 2 and Figure 5) may actually reduce gene expression for the majority of Cx-I-PEG5k, after normalization, it is revealed that the efficiency of gene expression increased with shorter alkyl chain lengths, except for C12-I-PEG5k (N/P = 16), contrary to the in vitro results (Appendix A). The gene expression efficiency of C12-I-PEG5k seems to be lower than that of C16-I-PEG5k; however, we consider the difference between C12-I-PEG5k (*P* = 0.07 vs. naked pDNA) and C16-I-PEG5k (*P* = 0.11 vs. naked pDNA) was not significant. Although the reason why pDNA alone exhibits such high expression levels upon direct intramuscular injection is unclear, physical pressure during the injection may be one of the factors for entering the cells in spite of its large size and lack of protection. The resulting in vivo gene transfection efficiency is presumably attributed to the following results: The shorter alkyl chains of Cx-I-PEG5k were more advantageous for pDNA release due to their moderate complex stability (Figure 8 and Appendix A) and lower toxicity (Figure 9 and Appendix A; pH 7 and Appendix A; right axis). Especially, C4-I-PEG5k had negligible cytotoxicity mediating higher transfection efficiency, as compared to C16-I-PEG5k mediating lower transfection efficiency, so that in vivo use of C4-I-PEG5k was also considered suitable and safe. The endosomal escape ability of the Cx-I-PEG5k alone (Appendix A: pH 5), presumably due to the proton (H^+^) buffering capacity around pH 6–7 by the alkyl imine group (Figure 4), and the enhancement of cellular uptake by the surfactant effect, contributed to the high transfection. Especially, the C4-I-PEG5k alone was hydrolyzed in one week (Appendix A), suggesting the enhanced dissociation of the I-MIC and the controlled release of pDNA. After two weeks, furthermore, the C4-I-PEG5k/pDNA I-MICs still mediated approximately twice the gene expression as compared to naked pDNA (Appendix A). The resulting long-term transfection efficiency may be due to the hydrolysis of the imine bond of the C4-I-PEG5k to decrease the stability of the I-MIC for the controlled relase of pDNA by dissociating PEG. The resulting dissociation of PEG by the imine bond would solve.

To examine further the in vivo gene transfection efficiency, as shown in Figure 11, the C4-I-PEG5k/pDNA I-MIC (N/P = 16), which exhibited the highest gene transfection efficiency (Figure 10B), was injected into mouse tibialis muscle in a similar method, followed by the visualization of the luciferase luminescence by in vivo imaging system (IVIS) after one week. In IVIS, the C4-I-PEG5k/pDNA I-MIC also exhibited higher gene transfection efficiency (Max value was 1.2 × 10^7^) (Figure 10B: fourth bar from the left), as compared to naked pDNA (Max value was 1.1 × 10^6^) (Figure 10B: the leftmost bar). Although IVIS images after two weeks are preferable for a time-dependent profile, from the viewpoint of gene expression area, notably, the C4-I-PEG5k/pDNA I-MIC seems to exhibit the luciferase luminescence in wider area, as compared to naked pDNA, suggesting diffusion into the tissues surrounding the tibialis muscle. The resulting gene expression in the wider area is considered a practical benefit of I-MIC. It should be noted that the resulting luciferase luminescence was observed only around the tibialis muscle, suggesting long-term safety without off-target effects. The presumable flexible structure of pDNA in the I-MIC (Figure 5 and Figure 6) is considered to be advantageous for diffusion-mediated penetration. The potential toxic effect of injecting both naked pDNA and I-MICs will be reported by assessing the induction of an inflammatory cytokine, such as TNF-α. As well, although the immunological response such as PEG-specific IgM should be evaluated, to minimize the IgM recognition, we have synthesized the C4-I-PEG5k by only terminal modification with a small hydrophobic group (butyl-imine), not hydrophobic long chains, to enhance IgM recognition [33]. Based on these results, although the C16-I-PEG5k/pDNA I-MIC below a N/P ratio of one might reduce cytotoxicity for good performance in biological assays, the C4-I-PEG5k/pDNA I-MIC led the highest gene transfection efficiency in vivo and is expected to provide local pDNA delivery with diffusion-mediated penetration.

## 4. Conclusions

We have been exploring pDNA delivery carriers useful for local skeletal muscles by altering the modifying groups of the PEG terminal structure. Cx-I-PEG5k, derived from a previous mono-cationic PEG, formed the I-MICs with pDNA, due to its high binding ability to pDNA, even though the positive charge of the carrier was less than that of a mono-cation. The ability to form the I-MICs depended on the length of the terminal alkyl chain; the longer alkyl chain promoted the formation of I-MICs, resulting in the higher stability of the I-MICs. Among the I-MICs, the C4-PEG5k/pDNA I-MIC at a mixing ratio (N/P) of 16 was less cytotoxic and preserved the flexible structure of naked pDNA, leading to the highest transfection efficiency in vivo. Consequently, the resulting I-MICs were stable against our previous MICs, offering the alkyl iminium for the pDNA binding group to demonstrate the factor of pDNA’s flexible structure as one of the key parameters for in vivo local pDNA transfection.

## Figures and Tables

**Figure 1 pharmaceutics-17-01054-f001:**
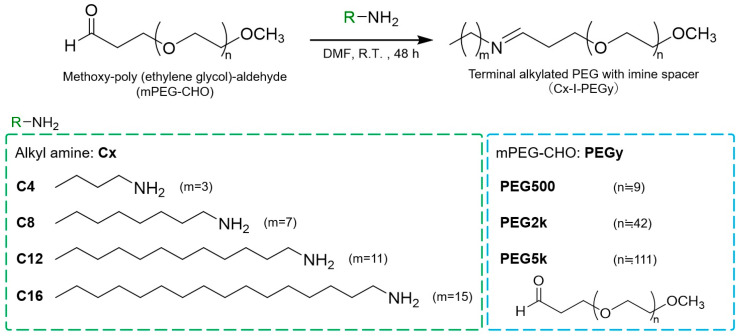
Synthesis scheme of terminal-alkylated PEG (Cx-I-PEGy), where *x* is the carbon number of each alkyl amine, and *y* is the molecular weight of PEG.

**Figure 2 pharmaceutics-17-01054-f002:**
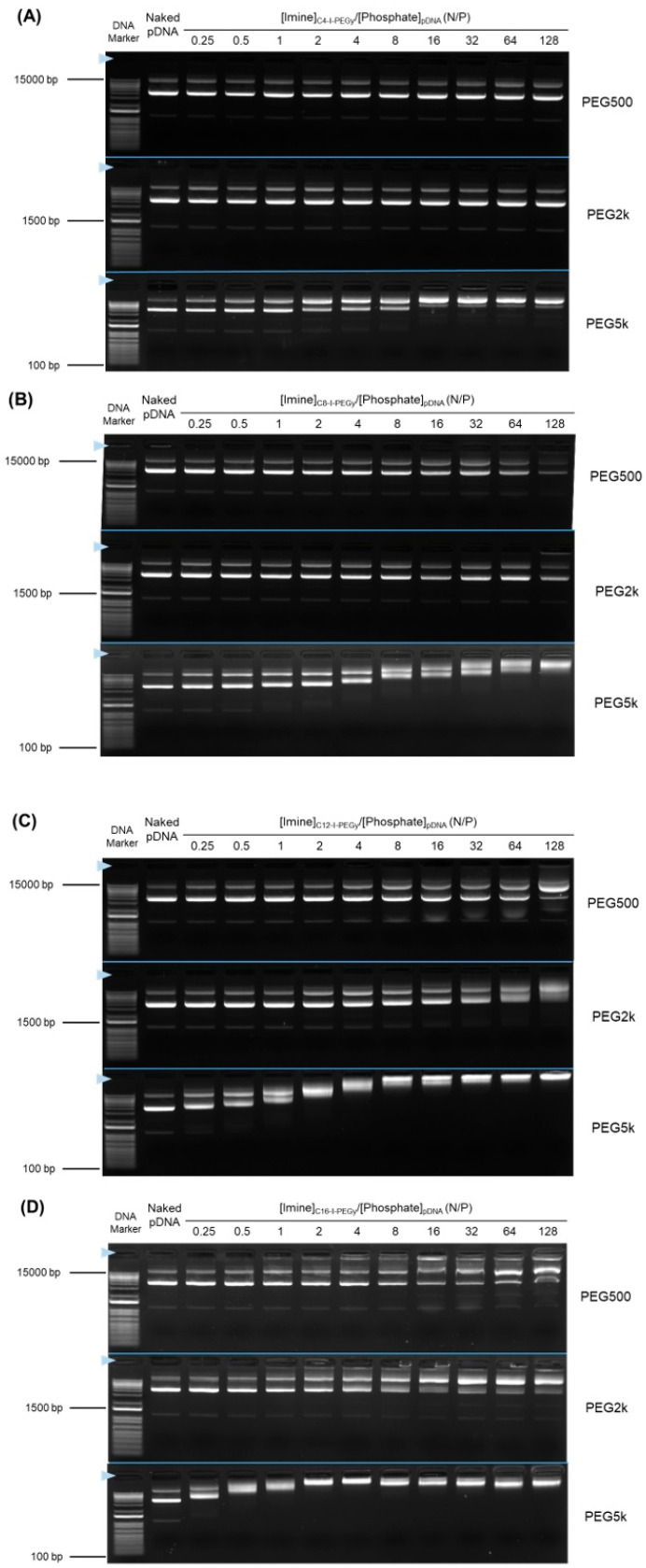
Complex formation between Cx-I-PEGy and pDNA as assessed by agarose gel electrophoresis. pDNA complex formation by (**A**) C4-I-PEGy, (**B**) C8-I-PEGy, (**C**) C12-I-PEGy, and (**D**) C16-I-PEGy, respectively. The mixing ratios of the imine of Cx-I-PEGy to phosphate group of pDNA, [Imine]_Cx-I-PEGy_/[Phosphate]_pDNA_ (N/P), are indicated. The blue triangle indicates the well where each sample was loaded.

**Figure 3 pharmaceutics-17-01054-f003:**
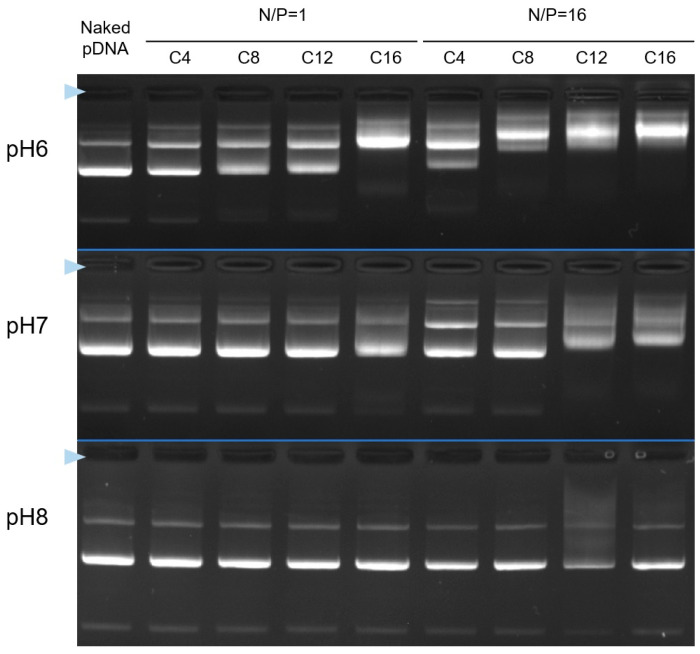
Formation of Cx-I-PEG5k/pDNA I-MICs under difference of pH as assessed by agarose gel electrophoresis. The complexes at mixing ratios (N/P) of 1 and 16 were incubated in phosphate buffer (PB) at pH 6, 7, and 8, respectively, and were electrophoresed in the same buffer at pH 6, 7, and 8. The blue triangle indicates the well where each sample was loaded.

**Figure 4 pharmaceutics-17-01054-f004:**
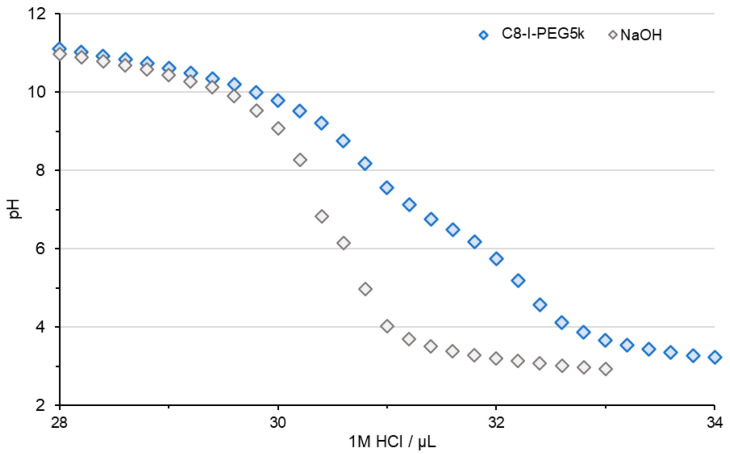
Acid–base titration curve of C8-I-PEG5k aqueous solution.

**Figure 5 pharmaceutics-17-01054-f005:**
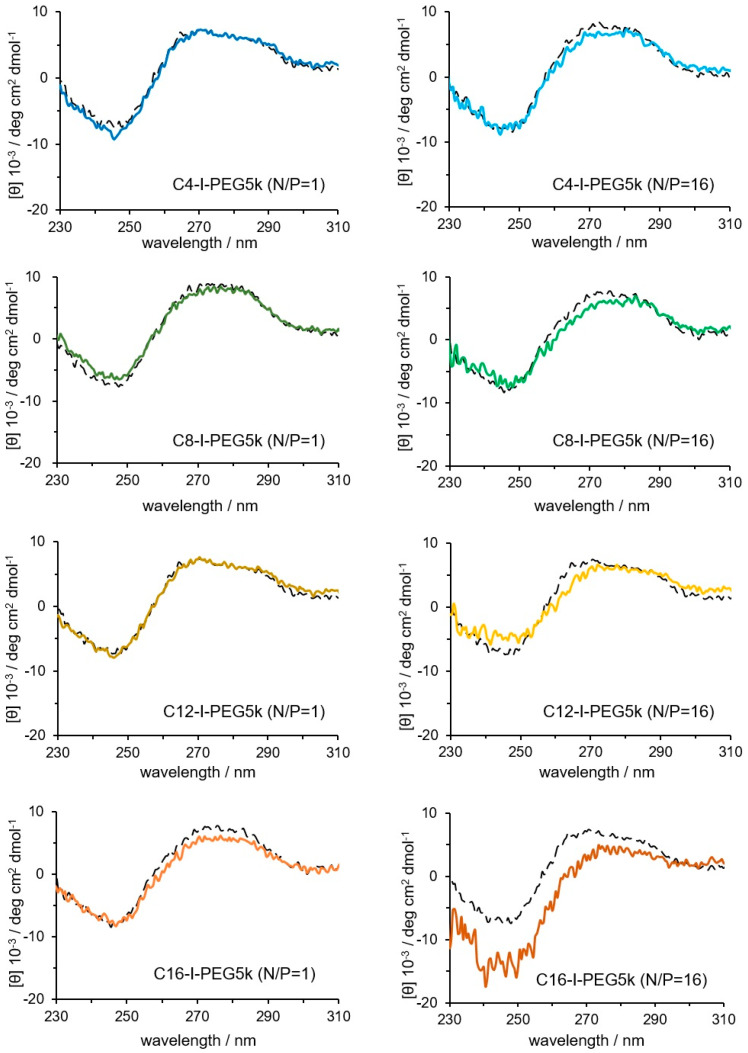
CD spectrum of the I-MICs between the pDNA and Cx-I-PEG5k at mixing ratios (N/P) of 1 and 16. The mixing charge ratios (N/P) of 1 and 16. The CD spectrum of the naked pDNA is represented as a black dotted line in each panel.

**Figure 6 pharmaceutics-17-01054-f006:**
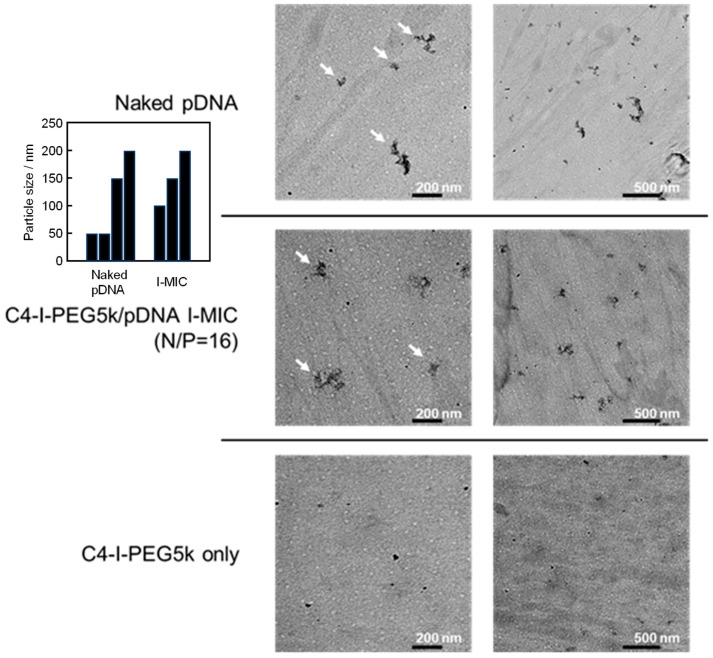
Representative TEM images of C4-I-PEG5k/pDNA I-MIC. The mixing ratio (N/P) was 16. Each left image is an enlargement of each right image. (Inset) Histogram of the size distribution indicated by arrows.

**Figure 7 pharmaceutics-17-01054-f007:**
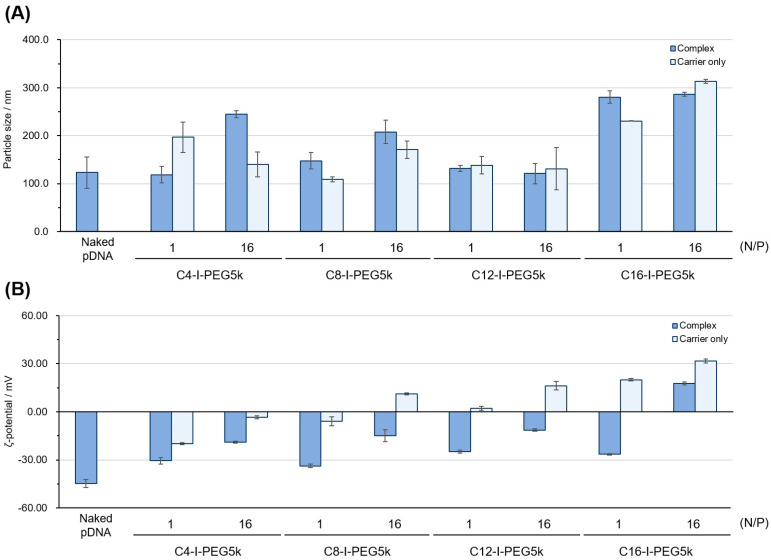
Particle size and zeta (ζ) potential of the Cx-I-PEG5k/pDNA I-MICs: (**A**) particle size and (**B**) ζ-potential. The mixing ratios (N/P) are 1 and 16. The dark blue bars indicate complex values, and the light blue bars indicate carrier only values. The error bars represent the mean and standard deviation of the measurements (particle size: *n* = 5, ζ-potential: *n* = 3).

**Figure 8 pharmaceutics-17-01054-f008:**
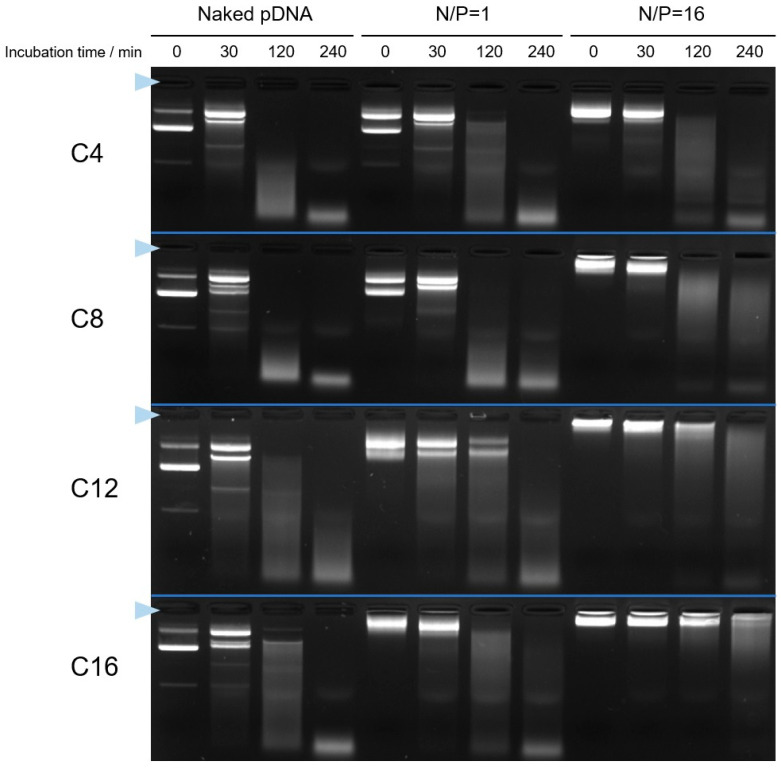
Stability of Cx-I-PEG5k/pDNA I-MICs in presence of serum protein as assessed by agarose gel electrophoresis. The complexes at mixing ratios (N/P) of 1 and 16 were incubated for 30–240 min in the presence of 10 % fatal bovine serum (FBS), followed by loading to the gel. The blue triangle indicates the well where each sample was loaded.

**Figure 9 pharmaceutics-17-01054-f009:**
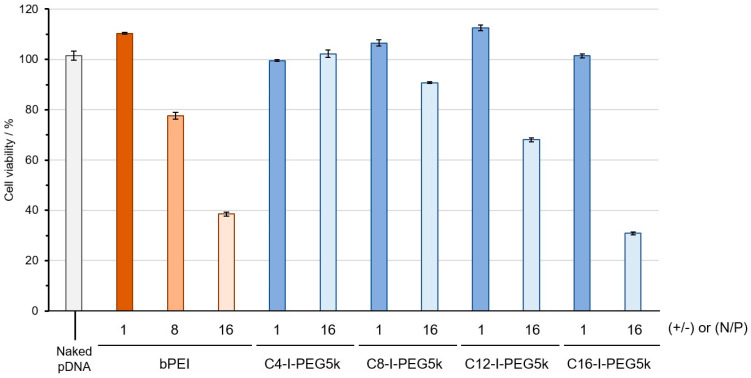
Viability of C2C12 cells in the presence of Cx-I-PEG5k/pDNA I-MICs and bPEI/pDNA complex at the mixing ratios (N/P) of 1 and 16 (for bPEI, the mixing charge ratios (+/−) of 1, 8, and 16). The vertical axis represents the percentage of viability relative to untreated control cells. The error bars represent the mean and standard deviation of the measurements (*n* = 3).

**Figure 10 pharmaceutics-17-01054-f010:**
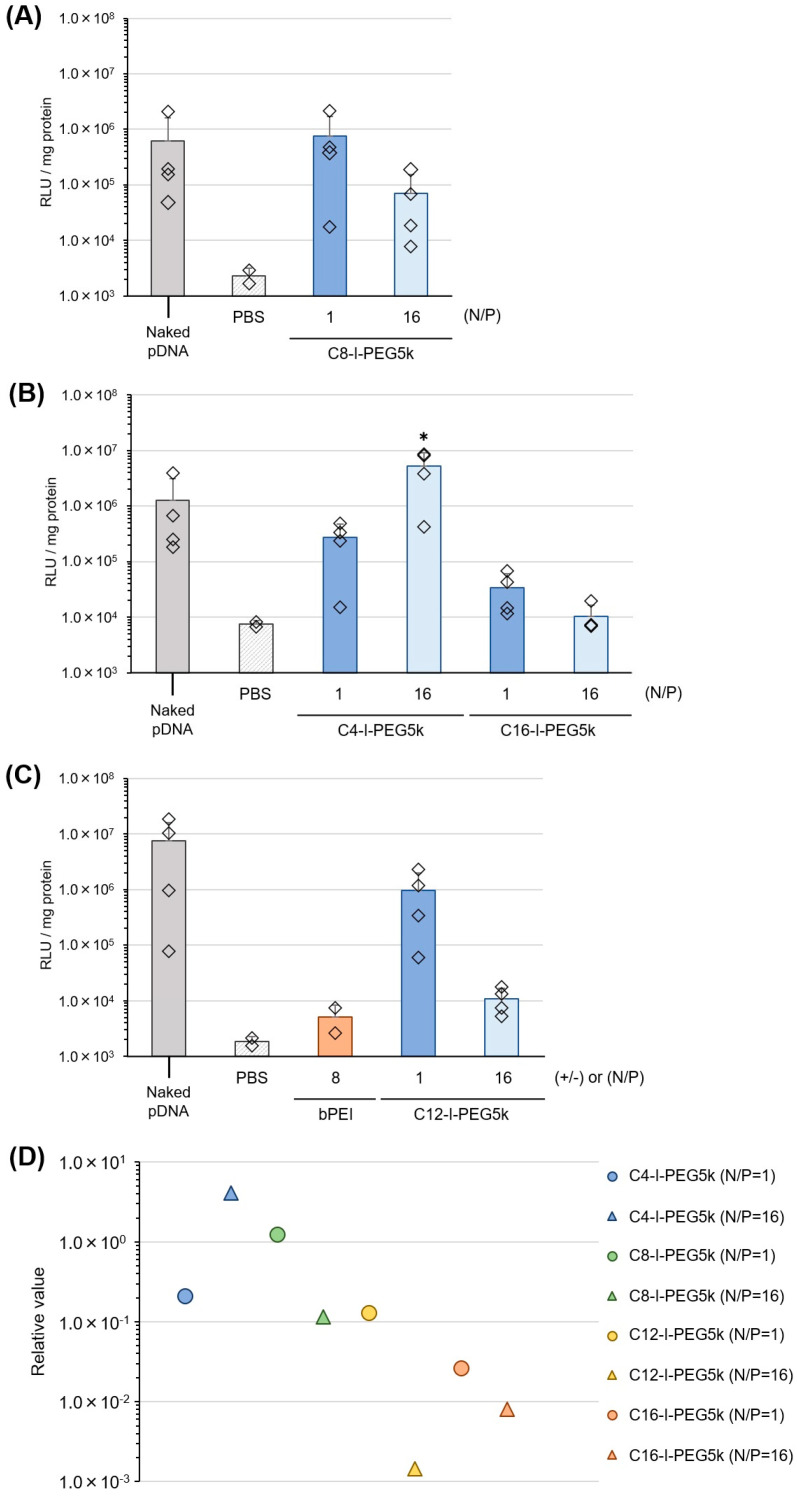
In vivo gene transfection efficiency in mouse skeletal muscles. Luciferase gene expression by intramuscular injection of pDNA complex with (**A**) C8-I-PEG5k, (**B**) C4-I-PEG5k and C16-I-PEG5k, and (**C**) C12-I-PEG5k and bPEI. The mixing ratios (N/P) are 1 and 16. Individual gene expression was determined via relative light unit (RLU) normalized by the protein concentration. Rhombic plots were indicated as an individual value. *p*-values are * *p* < 0.1 vs. naked pDNA. All data are represented as the mean and individual values (*n* = 2–4). (**D**) Gene expression of each complex was relative to gene expression of naked pDNA for each experiment. The circular plots indicate the value at the mixing ratio (N/P) of 1, and the triangular plots indicate the value at the mixing ratio (N/P) of 16.

**Figure 11 pharmaceutics-17-01054-f011:**
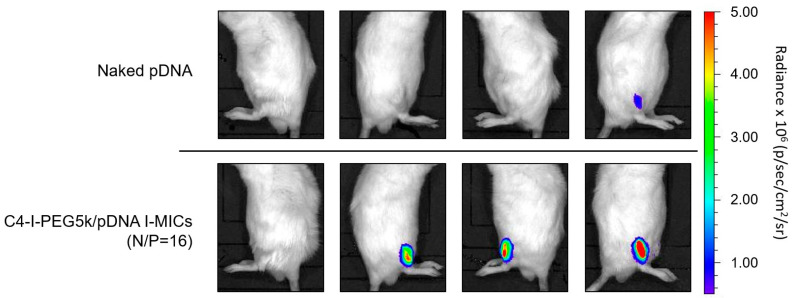
Representative observation images for in vivo gene expression: C4-I-PEG5k/pDNA I-MIC (N/P=16). Using an in vivo luciferase imaging system (IVIS), luciferase gene expression was observed at 1 week (*n* = 4). The color scale bar indicates radiance × 10^6^ (p/sec/cm^2^/sr) (Max = 5.0 × 10^6^, Min = 5.0 × 10^5^).

## Data Availability

The original contributions presented in the study are included in the article/Appendix A, further inquiries can be directed to the corresponding author/s.

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
