# Peer review of "Synthesis of Terminal-Alkylated PEGs with Imine Spacer to Form Iminium Mono-Ion Complexes for pDNA Delivery into Skeletal Muscle"

_pharmaceutics, 2025, doi:10.3390/pharmaceutics17081054_

Round 1
Reviewer 1 Report
Comments and Suggestions for Authors
The manuscript authored by Oba et al. describes the preparation of non-viral carriers for the delivery of plasmid DNA into skeletal muscle. In this regard, the authors include the synthesis of terminally alkylated PEG with a series of imine spacers, complex formation with pDNA, and full in vitro physicochemical characterization. Additionally, the authors evaluate biocompatibility and transfection efficiency in cell culture. Finally, an in vivo experiment is also discussed in the manuscript.
The article is interesting, and the topic appears to fit within the journal’s scope. Overall, the manuscript is well written and discussed. Although the authors demonstrate the potential of these non-viral vectors, I believe they should provide more in vivo evidence. Below are some comments and suggestions that should be addressed during the revision process:
- Page 3, line 99: Please add the volume of DMF used in the reaction.
- Page 3, line 105: Please include 13C-NMR and HR-MS data to fully characterize the PEGylated compounds.
- Page 3, line 110: Please evaluate the efficiency of plasmid incorporation into Cx-I-PEGy (EE%).
- Page 4, line 143: Please include the number of replicates performed.
- Page 4, line 163: Please include the ratios that have been evaluated in the assay.
- Page 7, line 243: Please correct the typo (“increaseing”).
- TEM images: Please include a histogram showing the size distribution of the particles.
- Figure 7: Please include data for untreated cells.
- Figure 9: Please include a time-dependent profile of luciferase expression quantified from IVIS images.
- The potential toxic effect of injecting both naked pDNA and complexes should be evaluated by assessing the induction of an inflammatory cytokine, such as TNF-alpha.
Author Response
Answers to Reviewer 1
Thank you for your kind reviewing.
- As your suggestion, we have added “ 2 mL of ” as the volume of DMF used in the reaction (p.3, line 110).
- As your suggestion, we carried out 13C-NMR and HR-MS However, it is difficult to detect a terminal small alkyl group (molecular weight of 15) in a huge PEG macromolecule (molecular weight of approximately 5000), so we have revise the text as follows (p.6, lines 242-247): Successful purification was confirmed by the disappearance of the proton signal due to the aldehyde of mPEG-CHO (signal CHO: Figure S1E) and the disappearance of the proton signal due to the methylene group adjacent to the primary amine of the unreacted alkylamine (signal g: Figure S1D). Although 13C-NMR and HR-MS data is preferable for full characterization, it is dificult to detect a terminal small alkyl group in a huge PEG macromolecule. Therefore, for each Cx-I-PEGy, successful synthesis was confirmed by the ratio of the proton signal due to the methoxy group of mPEG-CHO (signal a: 3H) to the proton signal due to the terminal methyl group (signal i: 3H), which was approximately 1:1.
- At a N/P ratio of 16, especially, no free pDNA band was observed in the presence of all Cx-I-PEG5k, suggesting 100% MIC formation. As your suggestion, we have added this in the revised text (p.8, lines 261-262).
- The number of replicates of DLS measurements was “ n = 5 ”, and that of ELS measurements was “ n = 3 ”. As your suggestion, we have added these in the revised text (p.4, lines 158 and 160).
- Especially, the cell viability at the mixing N/P ratio of 16 for C16-I-PEG5k was approximately 30%, loewer than bPEI. As your suggestion, we have added this in the revised text (p.15, lines 397-398).
- As your suggestion, we have corrected the typo (p.8, line 260): Replacement of “increaseing” with “increasing”.
- As your suggestion, we have included a histogram showing size distribution from TEM images (p.11, lines 322 and 323: Revised Figure 6).
- Thank you for your pointing out. The data in revised Figure 9 (original Figure 7) are represented against the data for untreated cells as 100%.
- As you point out, a time-dependent profile of luciferase expression quantified from IVIS images is preferable to include, as well as revised Figure S12 (luciferase expression after two weeks). So, we have added the clause “Although IVIS images after two weeks are preferable for a time-dependent profile,” in the revised text (p.17, line 462).
- As you point out, the potential toxic effect of injecting both naked pDNA and complexes should be evaluated. So, we have added the sentence “The potential toxic effect of injecting both naked pDNA and I-MICs will be reported by assessing the induction of an inflammatory cytokine, such as TNF-α.” in the revised text (p.17, lines 470-472).
Thank you for taking time from your busy schedule to review our manuscript.

Reviewer 2 Report
Comments and Suggestions for Authors
Dear authors have provided a solution for improving gene delivery with minimum abnormality in the structure of synthesized polyplex, but some issues shouldbe well clarified
1) Your developed method may not work in vivo, especially in IV injection mode
2) Endosomal escape is one of the main drawbacks for all the delivery modalities. How do you solve these issues with your method?
3) high positive charge and particle size ,especially ina 1:16 ratio have a huge impact on your therapeutic outcomes, please verify this problem
Author Response
Answers to Reviewer 2
Thank you for your kind reviewing.
1) Thank you for your suggestion. Our developed method is according to the approved method in Japan in 2019 by intramuscular injection (not IV). So, we have emphasized this in the revised text (p.1, line 31).
2) Thank you for your pointing out. Endosomal escape is presumable due to the proton (H+) buffering capacity around pH 6-7 by the alkyl imine group, as shown in revised Figures 4 and S9. We have added this in the revised text (p.16, line 444; p.17, line 445).
3) As your suggestion, we have added the discussion about the MIC at a high N/P ratio of 16 as follows (p.12, lines 344-346): Especially, in spite of a high N/P ratio of 16, the zeta potential of the C4-I-PEG5k/pDNA I-MICs was negative, suggesting the suitability of the I-MIC for in vivo therapeutic use.
Thank you for taking time from your busy schedule to review our manuscript.

Reviewer 3 Report
Comments and Suggestions for Authors
This manuscript describes application of alkylated PEGs for delivering DNA for gene therapy purposes. The manuscript is well-written, structured, and organized, the materials were characterized well, and it is easy to understand. This paper can be published after addressing a few questions, as follows:
- In the materials section, one of the PEGs is reported 550 Da, but in Fig. 1 is shown 500. Please clarify it.
- Why did the authors use bPEI? Why not linear PEI or lipofectamine, as a commercial and gold standard for in vitro transfection?
- The authors evaluated the toxicity of the polyplex; however, they need to evaluate the intrinsic toxicity of carriers without pDNA through the same assay.
- Why in Fig 8, naked pDNA has higher gene expression than the polyplex, while the luminescence signal in Fig 9 for naked pDNA is weaker (better so say no signal) than the polyplex?
- Most importantly: why did the authors choose this gene carrier which takes a long time to synthesis, purify, and characterize, while there are many biocompatible, simpler, and more user-friendly non-viral vectors? The reason is not clear and should be mentioned in the introduction. In fact, the authors must mention why they believe this the alkylated PEGs are better and more efficient than other non-viral carriers.
Author Response
Answers to Reviewer 3
Thank you for your kind reviewing.
・Thank you for your pointing out. Molecular weight of PEGy is presented as only PEG moiety without terminal aldehyde (-CH2CH2CHO = 57). So, the molecular weight is estimated to be approximately 500 (550-57 = 493). We have added this in the revised text (p.6, line 235).
・Thank you for your pointing out. The reason why we used bPEI is as follows: The bPEI is a polycation to from a PIC with higher cytotoxic cation density than linear PEI and lipofectamine is a lipid assembly, so we used bPEI as a positive control to compare I-MICs with a representative PIC. We have added this in the revised text (p.14, line 381-383).
・Thank you for your pointing out. The intrinsic cytotoxicity of 1 mM Cx-I-PEG5k is estimated from hemolytic assay at pH 7 (Figure S9). We have added this in the revised text (p.14, lines 393 and 394).
・Thank you for your pointing out. In Figure 9 (revised Figure 11), C4-I-PEG5k/pDNA I-MICs are used as a representative I-MIC mediating higher gene expression than naked pDNA in Figure 8 (revised Figure 10). The number of experiments is four (n=4). We have added this in the revised caption of Figure 11 (original Figure 9) for clear understanding (p.17).
・As your suggestion, we have mentioned why we believe this the alkylated PEGs are better and more efficient other non-viral carriers as follows (p.2, lines 81-87): The I-MIC is expected to have a nanoscale with its flexibility preserving pDNA structure by minimizing potential toxic cations, as compared to other non-viral carriers. One imine per carrier, because of not PIC but MIC, is weak base which is considered to work as bio-safe nonion under physiological pH without cytotoxic cation. When the I-MICs are formed, the imine is induced to be cationic iminium, expecting the minimization of the potential toxic cations under physiological pH.
Thank you for taking time from your busy schedule to review our manuscript.

Reviewer 4 Report
Comments and Suggestions for Authors
This study introduces an innovative approach utilizing terminal-alkylated PEGs with imine spacers for pDNA delivery, supported by promising in vivo gene transfection results. However, the study is constrained by limited statistical robustness, insufficient mechanistic and safety evaluations, incomplete long-term stability assessments, and a lack of comparative analysis with established delivery platforms. This is an original and novel study advancing pDNA delivery carrier design with potential biomedical applications.
However, authors should address the comments:
- An introduction statement is necessary at the beginning of the abstract. Authors should mention the method of characterization in the abstract. Explain about micelle, roles and applications.
- The selected keywords are complicated. Please revise them as scientific articles.
- Please revise and adjust writing, throughput of the manuscript by a native speaker.
- Revise text li lines 44-4 to a more clear sentences for readers.
- Are the Iminium Mono-Ion Complexes have a nanoscale? Please verify it (size measurement and morphology) and nanotechnology roles in the introduction. If the complex is in micelle form, size, size distribution, acual and hydrodynamic diameter is necessary to mention in abstract.
- Present the graphical abstract.
- Mention the gel retardation assay to confirm the complex formation.
- Extend a detailed method for animal study.
- Explain the software, statistics for comparison in section 2.16.
- Revise subtitle of 352, 380.
- Compare the transfection efficiency of the Peg conjugate with similar PEG based micelles in the literature.
- Explain hemolysis method in more details and mention the acceptance value based on ASTM. Hemolysis test is not standard and have not performed in different concentrations.
Addressing these limitations is crucial to enhance the translational potential of this delivery system:
- The manuscript focuses mainly on intramuscular gene delivery and expression efficiency but lacks detailed investigation or discussion regarding the biodistribution, clearance, and potential accumulation of Cx-I-PEGy polymers and pDNA-I-MIC complexes in vivo. This is critical for assessing long-term safety and off-target effects.
- In vivo experiments report gene expression measured in a relatively small number of animals (n=2-4). This small sample size limits the statistical power and confidence in the reported differences, especially given the noted wide inter-individual variability. Larger cohorts would strengthen conclusions.
- While three PEG molecular weights (500, 2k, 5k) were tested, the rationale for selecting these specific sizes is not strongly discussed. In particular, PEG500 and PEG2k showed poor complex formation, but manuscript could better explain why these MWs were chosen and how PEG chain length affects biodistribution/permeability beyond complex formation.
- Toxicity assays were restricted to in vitro cell viability (C2C12) and hemolysis. However, there is no in vivo toxicity assessment or systemic safety evaluation despite some formulations (e.g., C16-I-PEG5k at N/P=16) displaying notable cytotoxicity in vitro. Safety profiling is crucial before clinical translation.
- The complexes exhibit partial dissociation in physiological ionic strength solutions (PBS). Although authors suggest hydrophobic interactions mitigate this, the dynamic nature and possible premature release of pDNA in vivo under complex ionic environments remain underexplored. How about stability against DNase?
- Imine bonds are known to be hydrolytically labile, which is mentioned briefly with hydrolysis of C4-I-PEG5k over one week. However, detailed stability studies of the imine spacer under physiological conditions and their impact on transfection efficiency over time are limited.
- The observed variability and relatively modest fold increase (~2x) in gene expression compared to naked pDNA raises questions about the practical benefit of this system. Also, the reason why naked pDNA showed comparatively high gene expression after intramuscular injection is not fully clarified.
- Although some suggestions are made about surfactant effect and endosomal escape contributing to transfection, there is a lack of direct mechanistic experiments (e.g., cellular uptake quantification, endosomal escape assays) to support these claims.
- Besides bPEI in vitro, the manuscript does not include comparison to other clinically relevant gene delivery systems (e.g., lipid nanoparticles, viral vectors) in vivo to contextualize the performance of Cx-I-PEGy based I-MIC complexes.
- The immunological response to repeated intramuscular administration of these novel PEG-based carriers and pDNA complexes remains unknown and is not discussed.
- While stability in serum and salt is partially evaluated ex vivo, more comprehensive characterization (e.g., protein corona formation, aggregate size in serum) that influences in vivo behavior is missing.
- The challenge of the “PEG dilemma” (PEG reducing cell uptake) is only briefly mentioned. More detailed discussion or data showing how the imine spacer or alkyl modification overcomes this dilemma mechanistically would strengthen the manuscript.
- Some key data are relegated to Supplementary Materials (e.g., acid-base titration, TEM images). More prominent presentation or summary of these data in the main text could enhance understanding.
Best
Comments on the Quality of English LanguageExtensive English editing by a native speaker is necessaey.
Author Response
Answers to Reviewer 4
Thank you for your kind reviewing.
- As your suggestion, we have added “To design the pDNA delivery carrier into skeletal muscle” as an introduction statement (p.1, line 11), “as assessed by agarose gel electrophoresis” as the method of characterization (p.1, lines 15 and 16), and “without multivalent electrostatic interaction by minimizing potential toxic cations” as explanation about I-MICs (p.1, lines 16 and 17).
- As your suggestion, we have revised keywords such as plasmid DNA delivery, intramuscular injection, mono-ion complex, and alkyl imine (p.1, lines 25 and 26).
- As your suggestion, we have done as much as possible.
- As your suggestion, we have revised text as follows (p.2, lines 47-49): so that the mono-ion complexes [8] (MICs) with mono-cation PEG as an alternative pDNA delivery strategy to polycations have been designed for their flexible structure.
- Yes, the iminium mono-ion complexes (I-MICs) have a nanoscale. However, the I-MIC is not in micelle form. So, as your suggestion, we have revised the introduction as follows (p.2, lines 81-83; p.2, lines 87-89): The I-MIC is expected to have a nanoscale with its flexibility preserving pDNA structure by minimizing potential toxic cations, as compared to other non-viral carriers. The physicochemical properties were verified by dynamic light scattering (DLS) and circular dichroism (CD) measurement as well as transmission electron microscopy (TEM) observation.
- As your suggestion, we have already sent the graphical abstract to assistant editor (Ms. Lauretta Lou) as follows:

- As your suggestion, we have mentioned the gel retardation assay to confirm the I-MIC formation (p.3, line 121).
- As your suggestion, we have extended a detailed method for animal study as follows (p.6, lines 223 and 224): The Approval Code: P23-65, P24-53, P25-60. The Approval Date: May 7, 2024 (P25-60); May 16, 2024 (P23-65, P24-53).
- As your suggestion, we have explained the software, Microsoft Excel (p.6, line 227).
- As your suggestion, we have revised subtitle as follows: “Pharmaceutical Properties of Cx-I-PEG5k/pDNA I-MICs in vitro” (p.14, line 377) and “Pharmaceutical Properties of Cx-I-PEG5k/pDNA I-MICs in vitro” (p.15, line 413).
- As your suggestion, the comparison of the transfection efficiency of the PEG conjugate with similar PEG based micelles in the literature. However, as far as we know, there is no report of transfection efficiency mediated by PEG-based PIC micelles. We have added this in the revised text (p.15, lines 420 and 421).
- As your suggestion, according to the references 23 and 24, we have explained hemolysis method in more details as follows (p.5, lines 188 and 189): After centrifugation (13200 rpm, 10 min, 4°C), the amount of released hemoglobin from erythrocytes was determined by measuring the absorbance of the 5-fold diluted supernatant at 570 nm (n=6) from heme using a microplate reader [23,24].
- As your suggestion, we have revised the text in the viewpoint of long-term safety without off-target effects as follows (p.17, lines 466-468): It should be noted that the resulting luciferase luminescence was observed only around the tibialis muscle, suggesting long-term safety without off-target effects.
- As your suggestion, many animals should be used for in vivo Therefore, all number of animals used for Cx-I-PEG5k/pDNA I-MICs was four (n=4). We have added this in the revised text (p.15, line 419).
- As your suggeston, we have added explanation as follows (p.8, lines 253 and 254): Three PEG molecular weights (500, 2K, 5k) were tested, because 5k was used as a standard for biodistribution/permiability [7], and 500 and 2k were used as controls.
- As your suggestion, we have added discussion about in vivo safety as follows (p.15, lines 441-443): Especially, C4-I-PEG5k was negligible cytotoxicity mediating higher transfection efficiency, as compared to C16-I-PEG5k mediating lower transfection efficiency, so that in vivo use of C4-I-PEG5k was also considered suitable and safe.
- As your suggestion, we have described the stability as follows (p.13, line 367): as shown in revised Figure 8 (original Figure 6), we assessed the stability of the Cx-I-PEG5k/pDNA I-MICs in presence of serum proteins such as fetal bovine serum (FBS) including DNase.
- As your suggestion, we have added the related discussion as follows (p.17, lines 450-453): The resulting long-term transfection efficiency may be due to the hydrolysis of the imine bond of the C4-I-PEG5k to decrease the stability of the I-MIC for the controlled relase of pDNA by dissociating PEG.
- As your suggestion, we have added the discussion about the practical benefit of this system and the reason why naked pDNA showed high gene expression, with an additional reference 32, as follows: The resulting gene expression in wider area is considerd the practical benefit of the I-MIC (p.17, lines 465 and 466). Our previous report shows that the relationship between in vivo and in vitro gene transfection efficiency does not correlate [31], where the reason why nake pDNA showed comparatively high gene expression after intramuscular injection is unknown in spite of discussion [32] (p.15, lines 409-411).
- As your suggestion, we have added the discussion about cellular uptake and endosomal escape as follows (p.15, lines 404 and 405): Neither cellular uptake nor endosomal escape mediated gene transfection in vitro, alhough a direct mechanistic experiments are necessasy, so these results suggest that the surfactant effect of the Cx-I-PEG5k with longer alkyl chains disrupted the cell membrane (lower cell viability) to enhance cell uptake (higher gene transfection efficiency).
- As your suggestion, we have added the discussion about the comparison to other clinically relevant gene delivery systems as follows (p.15, lines 425-428): As a result, the C4-I-PEG5k/pDNA I-MIC (N/P=16) mediated the highest gene expression, as compared to other I-MICs and bPEI (representative PIC), where the comparison to other clinically relevent gene delivery sysytems, such as lipid nanopaeticles and viral vectors, was future scope.
- As your suggestion, we have added the discussion about the immunological response, with an additional reference 33, as follows (p.17, lines 472-475): As well, although the immunological response such as PEG-specific IgM should be evaluated, to minimize the IgM recognition, we have synthesized the C4-I-PEG5k by only teminal modification with a small hydrophobic group (butyl-imine), not hydrophobic long chain to enhance IgM recognition [33].
- As your suggestion, we have added the discussion about the stability in serum and salt as follows: Especially, at mixing ratios (N/P) of 16, almost no change of the pDNA bands in the I-MICs at 30 min suggests negligible protein corona formation and aggregation in serum (p.14, lines 368-370). Therefore, the formation of the Cx-I-PEG5k/pDNA I-MICs was assessed under physiological ionic strength conditions using PBS, because the stability of the I-MICs, formed by electrostatic interaction, depended on ionic strength (p.13, lines 350 and 351).
- As your suggestion, we have added the discussion about the challenge of the PEG dilemma as follows (p.17, lines 450-454): The resulting long-term transfection efficiency may be due to the hydrolysis of the imine bond of the C4-I-PEG5k to decrease the stability of the I-MIC for the controlled relase of pDNA by dissociating PEG. The resulting dissociation of PEG by the imine bond would solve the PEG dilemma as well as ester bond [13,14].
- As your suggestion, we have presented Figure S5 (acid-base titration) as Figure 4 in the main text (p.9). As well, we have presented Figure S7 (TEM images) as Figure 6 in the main text (p.11). So, we have sequentially revised figure numbers.
Thank you for taking time from your busy schedule to review our manuscript.

Round 2
Reviewer 1 Report
Comments and Suggestions for Authors
The authors have adressed all the reviewers' comments and therefore I recommend this manuscript for publication
Reviewer 4 Report
Comments and Suggestions for Authors.
Comments on the Quality of English LanguageMinor English editing is needed.